# Mangiferin Inhibits Apoptosis in Doxorubicin-Induced Vascular Endothelial Cells via the Nrf2 Signaling Pathway

**DOI:** 10.3390/ijms22084259

**Published:** 2021-04-20

**Authors:** Mohammad Bani Ismail, Peramaiyan Rajendran, Hamad Mohammed AbuZahra, Vishnu Priya Veeraraghavan

**Affiliations:** 1Department of Biological Sciences, College of Science, King Faisal University, Al Ahsa 31982, Saudi Arabia; habuzahra@kfu.edu.sa; 2Department of Biochemistry, Saveetha Dental College, Saveetha Institute of Medical and Technical Sciences, Saveetha University, Chennai 600 077, India; vishnupriya@saveetha.com

**Keywords:** mangiferin, doxorubicin, oxidative stress, Nrf2, reactive oxygen species, apoptosis

## Abstract

Doxorubicin increases endothelial permeability, hence increasing cardiomyocytes’ exposure to doxorubicin (DOX) and exposing myocytes to more immediate damage. Reactive oxygen species are major effector molecules of doxorubicin’s activity. Mangiferin (MGN) is a xanthone derivative that consists of C-glucosylxanthone with additional antioxidant properties. This particular study assessed the effects of MGN on DOX-induced cytotoxicity in human umbilical vein endothelial cells’ (HUVECs’) signaling networks. Mechanistically, MGN dramatically elevated Nrf2 expression at both the messenger RNA and protein levels through the upregulation of the PI3K/AKT pathway, leading to an increase in Nrf2-downstream genes. Cell apoptosis was assessed with a caspase-3 activity assay, transferase-mediated dUTP-fluorescein nick end labeling (TUNEL) staining was performed to assess DNA fragmentation, and protein expression was determined by Western blot analysis. DOX markedly increased the generation of reactive oxygen species, PARP, caspase-3, and TUNEL-positive cell numbers, but reduced the expression of Bcl-2 and antioxidants’ intracellular concentrations. These were effectively antagonized with MGN (20 μM), which led to HUVECs being protected against DOX-induced apoptosis, partly through the PI3K/AKT-mediated NRF2/HO-1 signaling pathway, which could theoretically protect the vessels from severe DOX toxicity.

## 1. Introduction

Doxorubicin (DOX) is an antineoplastic agent used for the treatment of several forms of cancers. Due to its amphoteric nature, doxorubicin travels to a variety of subcellular compartments and disrupts the integrity of intracellular nucleic acids, proteins, and lipid molecules [1]. These insults are not limited to cancer cells, and doxorubicin-induced cardiotoxicity is actually a product of the damage to noncancerous cells, especially cardiomyocytes [2,3,4]. As doxorubicin is given in the systemic circulation, the endothelium is the first cellular contact of the drug. Doxorubicin therapy will first affect the endothelial cells before it travels to other tissues such as the heart. Increased production of reactive oxygen species (ROS) and RNS can cause damage to mitochondrial DNA, proteins, and lipids, resulting in serious endothelial metabolic dysfunction [5,6,7]. A great amount of ROS could lead to increased calcium intracellular content and to disorders associated with calcium homeostasis, and could cause disorders of the FAS ligand transcription factor, leading to apoptosis [8]. Translocated nuclear factor erythroid 2-related factor (Nrf2) and transcription-mediated ARE (antioxidant response element) confer ROS-dependent cell death protection; the protective effects could be associated with the stimulation of Phosphoinositiden-3-Kinase (PI3K)/AKT signaling, which encourages cell survival [9]. Electrophile response elements (ARE) is also listed as the cis-acting transcription regulatory factor for gene activation, coding many antioxidant proteins’ quinone oxidoreductase (NQO1) and heme oxygenase-1(HO-1) are the ARE and phase II detoxifying proteins such as glutathione-S-transferases; [10,11]. Nrf2 is involved in stimulating the expression of both the ARE-regulating gene’s constitutive and inductive expression. In terms of the basal and inducible levels of antioxidant expression, there was an increase in oxidative stress, decreased antioxidant capacity in Nrf2 null mice, and impaired Nrf2/ARE function. This suggests that the NRF2/ARE pathway is essential for the regulation of intracellular redox status [12]. Doxorubicin has been reported to decrease the protein expression of B-Cell Lymphoma 2 (Bcl-2), which is usually known as a protein of interest and includes both pro-apoptotic and anti-apoptotic members. Thus, the oxidative and nitrosative stress initiated with doxorubicin cause mitochondrial damage and apoptosis (programmed cell death) in endothelial cells [13]. Therefore, endothelial cell-based anti-apoptotic strategies may minimize doxorubicin-mediated toxicity.

Mangiferin (MGN), a xanthone derivative and type of C-glucosylxanthone, is a monomer compound isolated from the plants of the Anacardiaceae and Gentianaceae families [14]. Mangiferin (1,3,6,7—tetra hydroxy xanthine -C2—beta -d-glucoside) has different pharmacological properties like anti-inflammatory, antioxidant, antitumor, and antidiabetic activity. In our earlier trials, we found that MGN is a very strong antioxidant [15,16]. Here we show that MGN activates Nrf2 signaling in DOX-induced endothelial cells.

## 2. Results

### 2.1. The Effect of Cisplatin and Mangiferin in HUVECs Is Dose- and Time-Dependent

A cell viability study with MTT was performed to assess the cytotoxic effect of MGN on human umbilical vein endothelial cells (HUVEC). Cell numbers after 12 and 24 h of incubation were not significantly affected by mangiferin (Figure 1B,C). Decreased cell viability (Figure 1D,E) was observed with 1 μM concentration of DOX: the cell viability was 58.3% (SEM ± 2%) greater than in control cells (Figure 1E) after 24 h. As shown in Figure 1F, treatment with DOX (1 μM) with MGN (10 or 20 μM) increased cell viability by more than 80% (20 μM). The cytotoxic effect of DOX with a protective effect of MGN was later verified and reconfirmed by the DNA fragmentation assay. Exposure to 1 μM DOX after 24 h of DNA fragmentation could be observed to have an effect in HUVECs (Figure 1G). These changes in cell fragmentation were, however, substantially attenuated in pretreated (20 μM) MGN cells. These findings clearly show that the exposure of HUVECs to MGN has an important protective effect against the vascular toxicity caused by DOX.

### 2.2. MGN Inhibits Late Apoptosis in DOX-Induced Vascular Endothelial Cells

Dual AO detection was used to analyze cell death modes [17]. AO proceeds to penetrate cells with damaged membranes, such as apoptotic and dead cells, and then emits bright green or red fluorescence into cells upon intercalation with DNA [17]. Earlier-stage apoptotic cells had bright green nuclei with fragmented chromatin and late-stage apoptotic cells had fragmented orange chromatin in the present study (Figure 1H).

### 2.3. Inhibition of NF-κB Activation by MGN

Activation of NF-κB can cause pro-inflammatory molecules to be overexpressed, which may lead to acute or chronic inflammatory disorders like atherosclerosis. Intracellular ROS accumulation has a direct correlation with mitochondrial dysfunction, pro-inflammatory cytokines’ induction via NF-κB nuclear translocation, and various pro-apoptotic cell signal cascades [18]. We explored whether the NF-κB activation could be inhibited by MGN. In comparison with that in control cells, DOX exposure considerably increased the NF-κB levels in the nuclear fraction of the vascular cells. The concurrent administration of MGN has been shown to be successful in restoring the nuclear NF-κB level (Figure 2A).

### 2.4. MGN Suppressed Pro-Inflammatory Cytokines Induced by DOX on HUVECs

Several other pro-inflammatory cytokines (COX-2 and TNF-α) were also identified as elevated in the DOX-administered cells. The administration of mangiferin was found to restore pro-inflammatory cytokines in vascular endothelial cells co-exposed to cisplatin and mangiferin (Figure 2A).

### 2.5. MGN Inhibits LDH Activity in DOX-Induced HUVECs

In Figure 2B, we can see that the LDH activity in the DOX group was significantly greater (*p* < 0.01) than that in the control group, suggesting that the endothelial damage was caused by DOX. However, the LDH activity in MGN with the DOX group declined significantly (*p* < 0.05), which indicates that MGN treatment could reduce the endothelial damage caused by DOX.

### 2.6. MGN Suppressed ROS Generation in HUVECs

The function of oxidative stress in DOX HUVECs was monitored by intracellular ROS generation. There was a significant rise in intracellular ROS with endothelial cells exposed to doxorubicin (DOX), as found using a DCFH-DA fluorescent probe (Figure 2C). However, the DOX-induced ROS production increase was significantly inhibited (*p* < 0.05) by MGN (10 or 20 μM) in a dose-dependent manner.

### 2.7. MGN Induces the Activation of Cell Survival Proteins

Various pathways caused by angiogenesis factors are shown to be involved in the angiogenesis process. The Phosphatylinositol 3-PIK3-kinase pathway is associated with VEGF and proliferation/expansion of basic fibroblast growth factor in endothelium survival and migration. Extracellular signals regulating the kinase [12] pathway, triggered by VEGF and FGF, have also been demonstrated in cell motility and cell survival regulation [20]. Hence, to demonstrate the involvement of pERK1/2/ERK in DOX with MGN-treated HUVECs, the phosphorylation of ERK1/2 was significantly induced 24 h in response to MGN with DOX-treated cells (Figure 3). Furthermore, the Western blot evidence supports a rise in the cell death pP38 protein in the DOX group and a decrease in DOX with MGN-treated cells (Figure 3). Initially, the changes in the levels of p-PI3K and p-AKT proteins were calculated by comparing the different groups. The levels of p-PI3K and p-AKT increased significantly in a dose-dependent manner following MGN incubation (20 μM) (Figure 3). The data showed that MGN triggered the proliferation of endothelial cells by activating the PI3K/AKT signaling pathway.

### 2.8. Activation of Nrf2 by MGN Attributed to PI3K/AKT Signaling Cascades in HUVECs

Nrf2 activation regulates PI3K/AKT signaling cascades. To identify the key signaling pathways involved in MGN-induced Nrf2 activation, HUVECs were treated with respective inhibitors of PI3K/AKT (LY294002) for 30 min, following treatment with MGN (20 μM) (Figure 4). PI3K/AKT inhibitors substantially suppressed MGN-induced Nrf2 activation in HUVECs. Therefore, MGN may rely on the regulation of PI3K/Akt-Nrf2 signaling in HUVECs.

### 2.9. MGN Upregulates HO-1 Expression in HUVECs through Nrf2 Activation

We hypothesized that the protective effects of MGN against DOX-induced oxidative stress are due to its induction of the transcription of antioxidant genes such as HO-1 and Nrf2. We observed that, as expected, MGN (10 or 20μM/mL) significantly increased the expression of cytoplasmic, nuclear, and total Nrf2 in a dose-dependent manner (Figure 5). The cells were pre-incubated with MGN in order to further examine the protective mechanism (10 or 20 μM/mL for 24 h), while oxidative stress was induced with DOX (1 μM/mL). The results of Western blotting demonstrated that the overall nuclear and cytoplasmic content of Nrf2 in DOX-induced cells decreased insignificantly and increased dramatically, respectively, as a result of MGN treatment after DOX exposure (Figure 5). In addition, the downstream target of Nrf2, HO-1 protein expression, in both stimulated and nonstimulated HUVECs, was significantly increased by MGN in a dose-dependent manner (Figure 5). In addition, DOX affected another NQO-1, while in MGN-treated cells we found MGN to reverse this expression. These data further support the fact that MGN pretreatment protects HUVECs from the oxidative stress caused by DOX as well as from DNA damage.

### 2.10. MGN Activates Nrf2 Translocation

Polyphenols’ ability to quench ROS is supplemented by increased antioxidant responses. In the presence of ROS, Nrf2, a redox-sensitive zipper protein, translocates into the nucleus, attaches to ARE in the promoter region, and initiates the transcription of a group of antioxidant genes. We examined the effects of MGN on Nrf2 activation in vascular endothelial cells since Nrf2 signaling is essential for apoptosis. Using confocal microscopy, the nuclear localization of Nrf2 in HUVECs with MGN (20 mol) was visualized. MGN treatment increased Nrf2 nuclear aggregation, as evidenced by the high Nrf2 staining in MGN-treated cells, according to immunofluorescence images (Figure 6A). To further confirm that MGN-induced Nrf2 activation is essential for preventing vascular endothelial damage, HO-1 and NQO-1 expression were determined in siNrf2-transfected cells, followed by MGN treatment. Western blotting showed that DOX-induced HO-1 and NQO-1 expression could not be prevented in siNrf2-transfected cells in the presence of MGN (Figure 6B).

### 2.11. MGN Activates Nrf2 Via AKT Signaling

We transfected HUVECs with AKTsiRNA to confirm this phenomenon; Nrf2 and HO-1 activation was seen in siNrf2-transfected cells, followed by MGN treatment. Western blotting showed that Nrf2 expression could be prevented in siAKT-transfected cells in the presence of MGN (Figure 6C). These results suggest that MGN would act on Nrf2 expression, modulated via AKT signaling.

### 2.12. MGN Upregulates the Bcl-2:BCl-xL Protein Ratio

To determine whether MGN activates Bcl-2 and Bcl-xL in HUVECs, we investigated their accumulation by Western blot analysis. We found that Bcl-2 and Bcl-xL levels increased in DOX with MGN therapy (10 or 20 μM/mL for 24 h) (Figure 7A). MGN could therefore activate Bcl-2 and Bcl-xL and suppress apoptosis.

### 2.13. MGN Downregulates Caspase3 and PARP Cleavage Proteins

The effect of MGN on downstream effector cascades, including caspase-3 and PARP, was then investigated. Western blot data showed that HUVECs with MGN and DOX-treated cells decreased caspase-3 levels dose-dependently (Figure 7A). Furthermore, cleavage in the PARP nuclear enzyme is triggered by the activation of caspase 3 and is a biological feature of apoptosis. As shown in Figure 7A, in response to the DOX-alone treatment, the PARP is cleaved to an 85-kDa fragment in HUVECs. Whereas in the case of MGN with DOX-treated cells, these effects were inhibited by MGN. These results indicate that MGN potentially inhibits DOX-induced apoptosis. Next, to further describe the mechanism underlying MGN to prevent the cell death caused by DOX, endothelial cells were treated with MGN (10 or 20 μM) and/or DOX (1 μM) for 24 h with DNA fragmentation, assessed by a transferase-mediated dUTP-fluorescein nick end labeling (TUNEL) assay. Images from fluorescence microscopy showed increased numbers of TUNEL-positive cells (green fluorescence) after treatment with DOX alone, while MGN treatment significantly reduced the number of TUNEL-positive cells (Figure 7B). These findings clearly show that MGN treatment prevents apoptosis in HUVECs induced by DOX.

### 2.14. MGN Inhibits Apoptosis, Regulated by the ROS Signaling Pathway in HUVECs

In this investigation, we pretreated HUVECs with ROS (NAC, 2 mmol) for 1 h, followed by MGN treatment (Figure 7C). Cells were reaped, and a Western blot was performed to check the signaling pathways engaged with MAN-actuated Bcl2 and downregulated cleaved caspase3 and PARP in HUVECs. MGN treatment appears to decrease the basal ROS levels in cells treated with DOX, and NAC co-treatment significantly inhibits ROS production, which reveals that MGN could inhibit apoptosis in HUVECs. These findings demonstrate that ROS signaling cascades are involving in the regulation of MGN-inhibited apoptosis by activated Nrf2 signaling.

## 3. Discussion

DOX is the drug of choice for the treatment of breast cancer, bladder cancer, and lymphocytic leukemia [21]. It is one of the most successful and safest therapeutic cancer drugs. However, DOX may induce irreversible cardiomyopathy in a dose-dependent manner, eventually resulting in heart failure [22,23]. The cardiotoxicity of DOX means that it has limited clinical use and is a main cause of noncancerous mortality [24]. In this research, we verified that DOX toxicity may cause significant endothelial vascular damage, as several functional or cellular indexes have demonstrated. Excess ROS generation is currently acknowledged to cause the subsequent pathophysiological shift in the cytotoxicity of DOX [25]. The pathophysiology of many human diseases, including vascular damage, includes cytokines, chemokines, and growth factors. High levels of pro-inflammatory cytokines were observed during cancer patients’ chemotherapy [26]. Regulating the balance of pro- and anti-inflammatory cytokines is an important part of resolving systemic side effects and chemotherapy toxicity [27]. As inflammatory cytokines have an effect on the protective role of the endothelial cells, various inflammatory cytokines have been measured in HUVECs [28]. Inflammatory cytokine (COX-2 and TNFα) expression has shown substantial growth in HUVECs with DOX-treated cells. Such an increase in inflammatory cytokines in HUVECs treated with DOX + MGN was not observed. These findings show that MGN protects against endothelial dysfunction in vitro through its potent anti-inflammatory action.

NF-κB serves as an essential mediator of inflammatory responses to various external and internal stimuli and mediates gene expression in cells. Our findings show that MGN prevents the DOX-induced translocation of NF-κB and its DNA binding. The activation of NF-κB in endothelial cells and cardiomyocytes has been documented to promote the pro-apoptotic role, while anti-apoptotic functions have been reported in cancer cells. Inhibition of NF-κB activation by MGN and the subsequent inhibition of target gene expression and inflammatory cytokines would also lead to MGN’s cardioprotective functions. For cell survival, the anti-apoptotic PI3K/AKT pathway is important. The PI3K pathway has been well established as an important signaling pathway for controlling cell growth, proliferation, survival, and migration. A crucial mediator for the survival of the cell is the viral oncogene *homolog1*, with AKT triggered in response to growth factors, Ca2+ influx, and oxidative stress. A wide variety of polyphenols, including kaempferol, pinocembrin, and DATS, have been reported in pathways dependent on PI3K/AKT/Nrf2 for oxidative cell damage [10,29,30]. PI3K/molecular AKTs’ mechanical pathway begins with the extracellular oxidative stress that activates tyrosine kinase in the cell surface. Once the PI3K is activated and activates the anti-apoptotic kinase, leading to Nrf2 phosphorylation from its suppressor Keap1 gene, it facilitates Nrf2 translocation into the nucleus and thus promotes phase II antioxidant enzymes. In the present work, we found a reduction in PI3K/AKT and Nrf2 proteins in DOX relative to DOX with MGN. Pretreatment with MGN significantly increased PI3K and AKT, as demonstrated by Western blotting in MGN-treated HUVECs. The findings of this study are further confirmed by the endogenous antioxidant upregulation through the activation of Nrf2 via the PI3/AKT and ERK pathways, which can prevent endothelial damage.

Nrf2 controls the expression of many ROS detoxifying and antioxidant genes. Nrf2 interruption arrests the G2-M cell cycle by redox signals caused by impaired GSH in the cultivated primary epithelial cells. A recent study showed that a bone marrow-derived cell deficiency of Nrf2 aggravates atherosclerosis in LDLR mice [31]. Different polyphenols, like kaempferol, curcumin, quercetin, and resveratrol, have been shown to provide endothelial defense or defend against Nrf2 induction in atherosclerosis [32,33]. Nrf2 was also described as a possible vascular dysfunction, as opposed to these recorded protective actions. Our data support previous findings on Nrf2 protective measures and indicate that MGN could activate the Nrf2/HO-1 pathway, providing protection against endothelial dysfunction and atherosclerosis. In this study, we discovered a substantial increase in the protein expression of Nrf2 and HO-1 from DOX with MGN in HUVECs. HO-1 is known to play an important protective function in endothelial vascular disease. This positive effect is also linked to improved artery protection against the endothelial dysfunction caused by oxidants such as HOCl. Arterial induction of HO-1 could protect nitric oxide synthase against oxidative damage, while increased NO can induce HO-1. Endothelial function was rescued by MGN in HUVECs exposed to DOX.

The mammalian apoptosis pathway is regulated by Bcl-2 protein groups, including Bcl-2, Bax, Bak, and Bcl-xL. Bax, a proapoptotic member of the Bcl-2 family, translocates from the cytoplasm to the mitochondria, where it induces disturbances in mitochondrial activity, releases cytochrome c, and contributes to apoptosis [34]. Another anti-apoptotic protein, Bcl-2, prevents the release of cytochrome c and mediates anti-apoptotic effects. DOX can cause p53 activation, resulting in p53-dependent downregulation of Bcl-2 in the apoptosis of endothelial cells [35]. The present study has nevertheless determined that MGN has considerably enhanced the inhibitory effects of DOX on expression of Bcl-2 in HUVECs, possibly by p53 inhibition. In addition, Sahara et al. stated that MGN upregulated the Bcl-2 expression in a concentration-dependent way. DOX enables the apoptotic signaling pathway, as shown by proapoptotic protein upregulation with antiapoptotic protein downregulation. MGN can downregulate Bax overexpression induced by hypoxia and simultaneously upregulate Bcl2 expression reduced by hypoxia. Furthermore, MGN demonstrates anti-apoptotic signaling cascade behavior by opening up mKATP channels and enhancing their nitrate-like effect to prevent the loss of the mitochondrial membrane potential caused by ROS. Our research has shown that after the addition of MGN, TUNEL-positive, DOX-treated HUVECs were significantly reduced. We noted that caspase-3 activity increased significantly with DOX-treated cells and MGN significantly decreased the activation and increased the expression of Bcl-2 in HUVECs. Further research is needed for the complete clarification of the molecular mechanisms by which MGN has mitochondrial antiapoptotic effect.

## 4. Materials and METHODS

### 4.1. Reagents

The reagent contained mangiferin, doxorubicin, dimethyl sulfoxide (DMSO), SDS, tris, glycine, BSA, 3-(4,5-dimethythiazol-2-yl)-2,5-diphenylTrazolium Bromide (MTT), and CelLytic (Sigma-Aldrich, St. Louis, MO, USA). Other ingredients included Dulbecco’s Modified Eagle Medium (DMEM), fetal bovine serum (FBS), antimycotic/antibiotic mixtures (Carlsbad, CA, USA), DCFH2-DA, and TUNEL staining kits (Roche, Mannheim, Germany), and antibodies were purchased from Invitrogen, Carlsbad, CA, USA. Antibodies to AKT, pPI3K, PI3K PERK1/2, ERK1/2, pP38, P38, BCL2 PARP, HO-1, NQO1, Keap-1, pNrf2, COX, TNF-α, pNFC-B, β-actin, and Lamin B1 were sourced from Invitrogen and Thermo Fisher, Waltham, MA, USA. The antibody against cleaved caspase-3 was purchased from Abcam (Branford, CT, USA).

### 4.2. Cell Culture

HUVECs were obtained from the American Type Culture Collection (Manassas, VA, USA) and cultured at 37 °C with 5% CO_2_ in DMEM/F12 medium containing 10% FBS (HyClone, Logan City, UT, USA).

### 4.3. Treatment of Cell Culture

The serum-free medium was suspended with MGN and DOX. Cells were seeded on the plates and then added in triplicate over 24 h in each group with MGN, DOX, or a combination of the two. Cells were added to a 0.25% trypsin (w/v) and 0.52 mmol EDTA solution (Thermo, Waltham, MA, USA).

### 4.4. Viability Assay

A MTT test was used to assess the viability of cells. These cells were then kept in 12-well plates at 4‒105 cells per well, and subjected to different treatments. The cells were then grown for 2 h in a medium containing 400 μL of MTT solution (0.5 mg/mL). Once the medium was removed, the product was liquefied with DMSO (400 μL). Lastly, the absorption of each well was calculated by a microplate reader (Bio Tek, Winooski, VT, USA), at 570 nm. Untreated cells were used as a control for monitoring cell viability.

### 4.5. Detection of Apoptosis, DNA Fragmentation, and ELISA

Cellular DNA fragmentations were detected using a cell death detection ELISAPLUS package (Roche Molecular Biochemicals, Mannheim, Germany) according to the manufacturer’s protocol, as mentioned in previous studies [36].

### 4.6. In Situ Detection of Apoptosis

After 24 h, fluorescent staining methods quantified apoptosis via the identification of AO condensed nuclei by a fluorescent microscope. AO staining distinguishes between living and apoptotic cells. Subconfluent (70‒80% confluent) cells were subjected to apoptotic agents for 24 h. In 24-well plates, HUVECs were gently washed with PBS and immediately treated with acridine orange (100 μg/mL) and ethidium bromide (100 μg/mL) for 5 min. Both wells were inspected under an epifluorescence microscope. The exact wavelength used to test fluorescence microscopy was 455 nm. The morphological criteria describe apoptosis.

### 4.7. Determination of Lactate Dehydrogenase Activity

In HUVECs, LDH is an intracellular enzyme released to the culture medium during cell damage [37]. The solution was obtained at the end of the experiment and the LDH activity was assayed according to the requirements of the LDH test kit by a microplate reader (Biorad 680, Jiancheng, Nanjing, China).

### 4.8. Isolation of Nuclear and Cytoplasmic Fraction

This procedure has been performed according to standardized methods. Cells were dissociated in 1× hypotonic buffer and collected. Twenty-five microliters of detergent were added and the supernatant cytoplasmic fraction was collected at 14,000× *g* after centrifugation. The nuclear pellet was resuspended by the lysis buffer. The nuclear fraction supernatant was stored at −80 °C after vortexing and centrifugation.

### 4.9. Western Blotting Analysis

Western blot analysis was performed as previously discussed [38]. In a nutshell, cell lysates were separated and transmitted to the nitrocellulose membrane via SDS–PAGE. Different antibodies blocked and probed the membrane. The proteins were identified with enhanced chemiluminescence (Millipore, Bedford, MA, USA). The bands were quantified with ImageJ software (v. 1.8.0, National Institutes of Health, Bethesda, MD, USA).

### 4.10. Immunofluorescence

Immunofluorescence was performed according to a procedure previously described [39].

### 4.11. siRNA Transfections

siRNA transfection was carried out as described earlier [40]. To determine whether MGN interferes with ROS through modulating Nrf2 signaling, HUVECs were transfected with Nrf2 siRNA and the control scrambled with a transfection reagent.

### 4.12. Reactive Oxygen Species Generation Assay

Fluorescence microscopy was used to detect DCF fluorescence in cell tissue, demonstrating the presence of intracellular ROS. The cells (5 × 10^4^ cells/mL) were grown in DMEM/F12 with 10% FBS. The culture medium was changed when the cells reached 80% confluence. Following treatment with MGN + DOX, the cells were washed in PBS and then incubated in fresh culture medium with 10 μM of DCFH2-DA. An intracellular esterase separated the DCFH2-DA acetate groups and trapped the probe in AGS cells. Intracellular ROS were measured with a DCF fluorescence microscope.

### 4.13. TUNEL Assay for Apoptotic DNA Fragmentation

As described above, DNA fragmentation was assessed using deoxynucleotidyl oligonucleotide TUNEL assays. After treatment with FKB and/or doxorubicin, the apoptotic cells (2 × 10^4^ cells/mL in eight-well plates) were collected and 4% formaldehyde was used for fixation and mounting. TBS-fixed cells were permeable to 20 g/mL of proteinase K and 3% hydrogen peroxide was used with endogenous peroxidase in methanol. The 3′OH ends of DNA with biotin-dNTP were observed to measure apoptosis using the Klenow enzyme at 37 °C for 1.5 h. These slides were then incubated with streptavidin combined with 3,3′-diaminobenzidine, hydrogen peroxide, and horseradish peroxidase. Their fluorescent nuclei were used to classify the fragmented DNA under a fluorescence microscope.

### 4.14. Statistical Analyses

Data were represented in normal distribution as the mean ± standard error of mean (SEM) and one-way analysis of variance, and a Tukey post hoc analysis was used to analyze the differences between groups. The survival rates were evaluated by the Kaplan‒Meier method in the four different groups of mice. The statistical significance of the study was set to *p* < 0.05.

## 5. Conclusions

In conclusion, we demonstrated that MGN may have prophylactic and therapeutic efficacy in protecting against DOX-related oxidative endothelial damage due to its potent antioxidant capacity. By activating the Nrf2 pathway via PI3K/AKT, MGN counteracted DOX-induced toxicity in the endothelium. This later increased Nrf2-dependent antioxidant signaling and restrained the inflammatory response and apoptosis. Nrf2 has been a much-discussed gene of inquiry for large gene that led to decreased vascular oxidative damage caused by DOX. In the near future, further comprehensive studies may identify MGN as a bioactive candidate for the treatment of DOX-associated vascular complications.

## Figures and Tables

**Figure 1 ijms-22-04259-f001:**
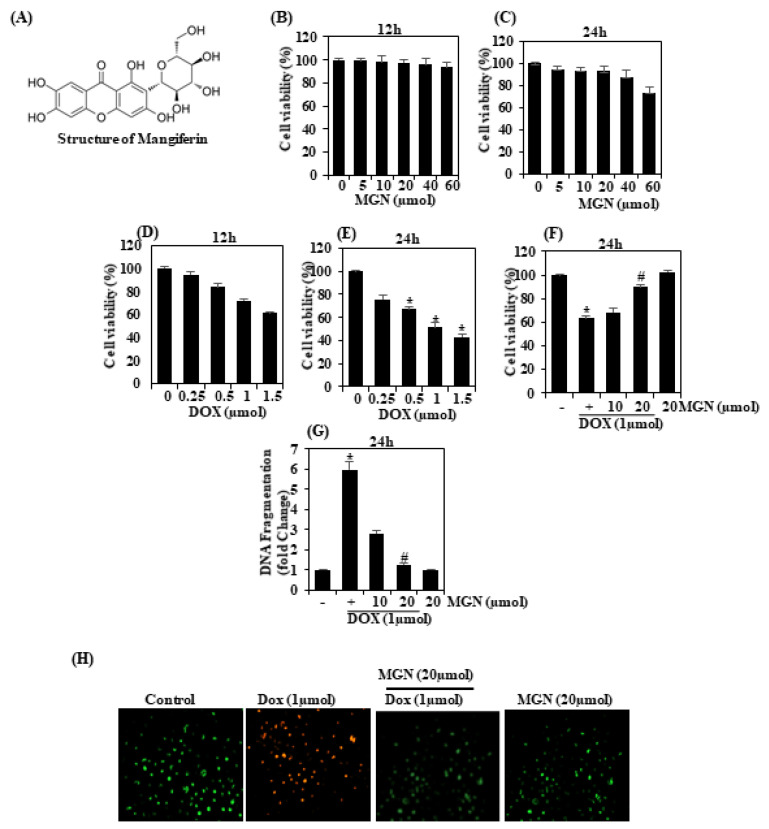
The effects of doxorubicin (1 µM), mangiferin (MGN, 10 or 20 µM) or a combination thereof on the proliferation of HUVECs. (**A**) Chemical structure of mangiferin. (**B**,**C**) MGN on HUVECs’ viability (0‒60 μM for 12 or 24 h, respectively). (**D**,**E**) Cell viability effect of DOX on HUVECs (0‒1.5 μM for 12 or 24 h, respectively). (**F**) MGN (10 or 20 μM) cytoprotective on DOX (1 μM)-induced HUVECs for 24 h. (**G**) Effect of MGN on DNA fragmentation in DOX-induced endothelial cells. (**H**) Acridine orange/ethidium bromide (AO/EB) staining observed for cell apoptosis for 24 h. The outcome was measured and expressed as a fold change compared to the control (*n* = 3). Data were analyzed by one-way ANOVA, and Tukey’s post hoc experiments were used to analyze the similarities between groups. * *p* < 0.05 reflects significant differences from the healthy control. # *p* < 0.05 reflects significant differences in contrast to DOX alone and MGN with DOX-treated groups.

**Figure 2 ijms-22-04259-f002:**
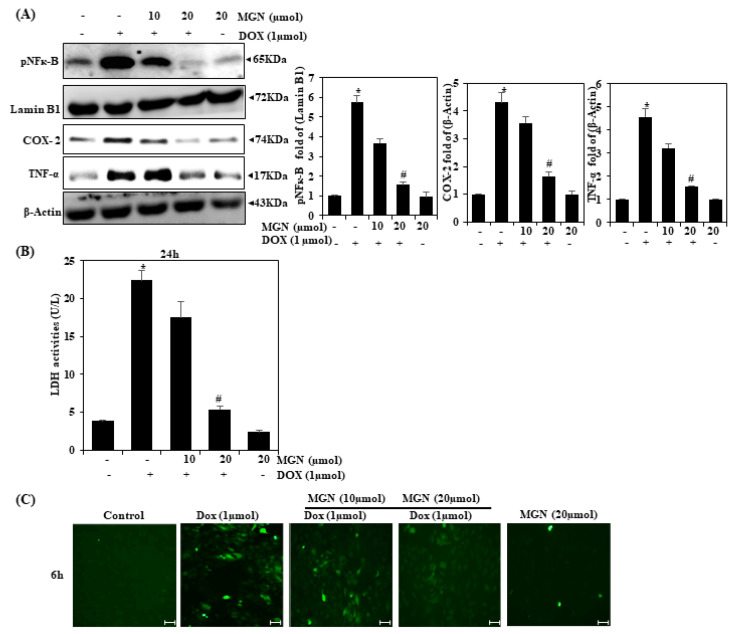
Effect of DOX and mangiferin on pro-inflammatory cytokines in HUVECs. (**A**) pNFкB, COX-2, and TNFα activation of MGN and/or DOX in HUVECs. The bands were quantified using Image Studio Lite software, and the differences are represented by a histogram. The internal controls used were Lamin B1 and β-Actin. (**B**) Lactate dehydrogenase [19] development in endothelial cells treated with MGN. (**C**) In the DCFH-DA assay, a nonfluorescent membrane permeable probe was applied to the culture medium 30 min before the end of each experiment. DCFH-DA metabolized fluorescent DCF with cellular ROS. The outcome was measured and expressed as a fold change compared to the control (*n* = 3). Data are represented as mean ± SEM. One-way ANOVA plus Tukey’s post hoc analysis analyzed the comparison between groups. * *p* < 0.05 reflects significant differences from the healthy control. # *p* < 0.05 reflects significant differences in contrast to DOX alone and MGN with DOX-treated groups.

**Figure 3 ijms-22-04259-f003:**
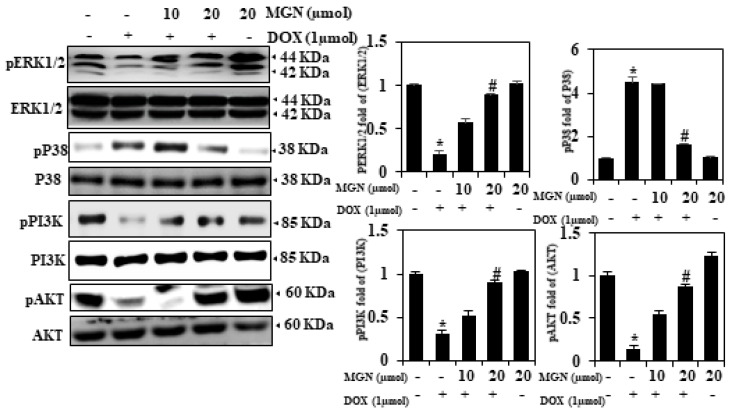
MGN’s influence on cell survival proteins. Cells have been treated with MGN and/or DOX (10 or 20 μM) for 24 h. Whole-cell lysate was examined by pERK, p-PI3K, pP38, or p-AKT in Western blot. Total ERK, P38, AKT, and PI3K levels were evaluated as loading controls. The outcome was measured and expressed as a fold change compared to the control (*n* = 3). Data are mean ± SEM. One-way ANOVA plus Tukey’s post hoc analysis analyzed the comparison between groups. * *p* < 0.05 reflects significant differences from the healthy control. # *p* < 0.05 reflects significant differences in contrast to DOX alone or MGN with DOX-treated groups.

**Figure 4 ijms-22-04259-f004:**
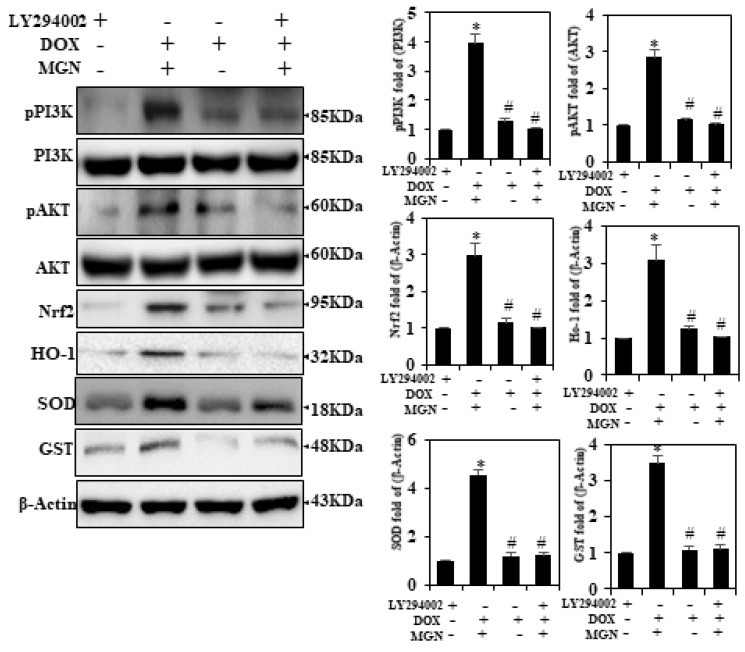
MGN-mediated Nrf2 activation by PI3K/AKT signaling pathways in HUVECs. Cells were pretreated with PI3K/AKT inhibitor LY294002 (30 μM) for 30 min, followed by MGN (20 μM) and/or DOX (1 μM) for 24 h. Western blot was performed to detect the pPI3K, pAkt, HO-1, Nrf2, SOD, and GST levels by anti-pPI3K, anti-pAkt, anti-HO-1, anti-Nrf2, anti-SOD, and anti-GST. The outcome was measured and expressed as a fold change compared to the control (n = 3). Data were analyzed by one-way ANOVA, and Tukey’s post hoc experiments were used to analyze the similarities between groups. * *p* < 0.05 reflects significant differences from the healthy control. # *p* < 0.05 reflects significant differences in contrast to DOX alone and MGN with DOX-treated groups.

**Figure 5 ijms-22-04259-f005:**
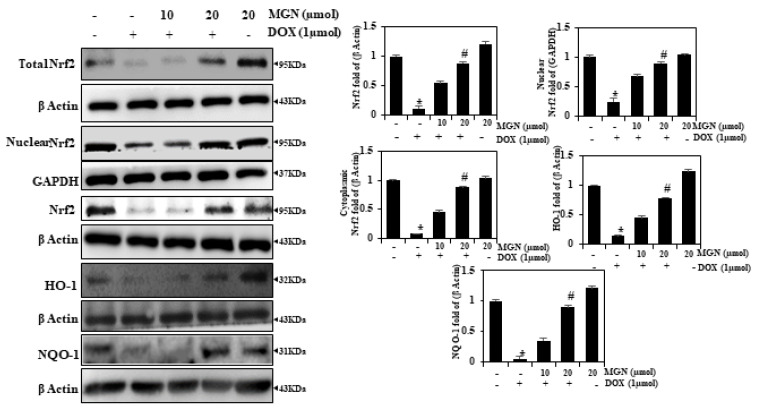
MGN upregulates the antioxidant biomarkers Nrf2, NQO1, and HO-1 in DOX-induced endothelial cells. MGN (10 or 20 μM for 24 h) and/or DOX were used for the pretreatment of cells. Western blot findings showed the effects of MGN on Nrf2, NQO-1, and HO-1 on cytoplasmic, nuclear, and total protein concentrations in whole cells. For each sample, 50 μg of lysate were resolved in total, with β-actin used as an internal control protein. Densitometry was conducted to calculate relative shifts in protein band intensities. The outcome was measured and expressed as a fold change compared to the control (*n* = 3). Data are represented as mean ± SEM. One-way ANOVA plus Tukey’s post hoc analysis analyzed the comparison between groups. * *p* < 0.05 reflects significant differences from the healthy control. # *p* < 0.05 reflects significant differences in contrast to DOX alone and MGN with DOX-treated groups.

**Figure 6 ijms-22-04259-f006:**
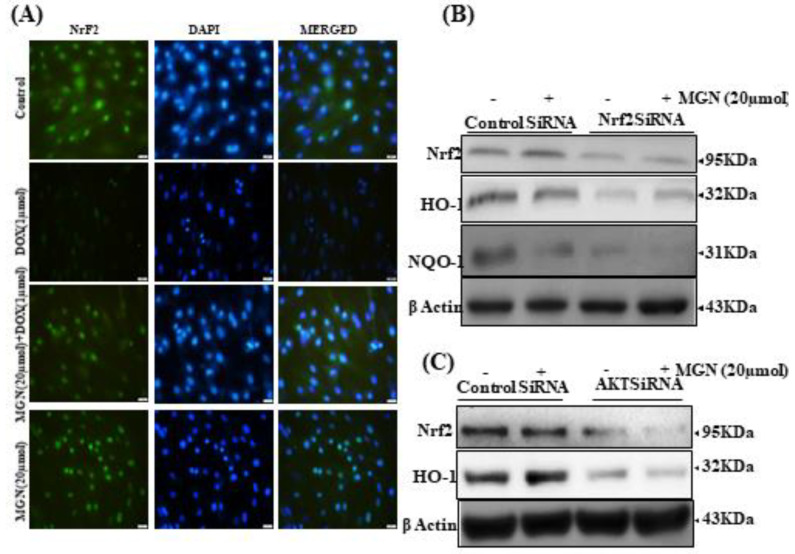
Pretreatment with MGN protects endothelial cells from DOX-induced apoptosis. For the study, the control HUVECs and pretreated cells (20 μM MGN for 24 h) and/or DOX. (**A**) Immunofluorescence analysis (200× magnification) was used to evaluate the nuclear translocated Nrf2 expression. (**B**) HUVECS were transfected with a specific siRNA against Nrf2 or a control siRNA. Western blot analysis measured Nrf2, HO-1, and NQO-1 protein levels in cells transfected with control siRNA or siNrf2. (**C**) HUVECs were transfected with a specific siRNA against AKT or a control siRNA. The Western blot analysis measured Nrf2 and HO-1 (24 h) protein levels in cells transfected with control siRNA or siAKT. HUVECs were treated with MGN (20 μM) for the indicated time.

**Figure 7 ijms-22-04259-f007:**
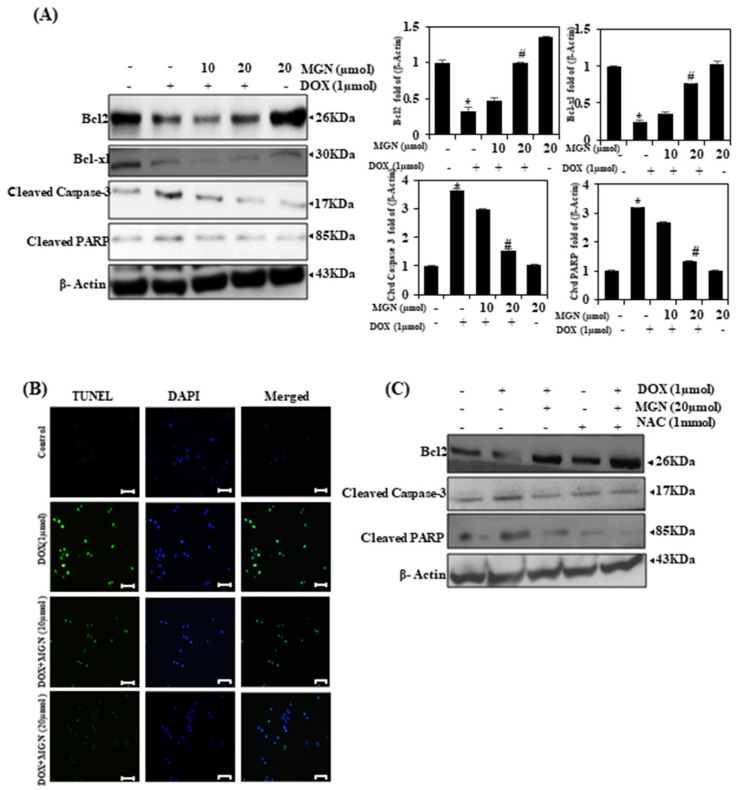
Pretreatment with MGN protects endothelium cells from DOX-induced apoptosis. For the study, control HUVECs and pretreated cells (10 or 20 μM MGN for 24 h) and/or DOX. (**A**) Expression level was assessed with immune blotting for apoptotic regulatory proteins such as Bcl-2, Bcl-xL, caspase-3, and PARP. SDS–PAGE is used as a routine tool for protein purification, with β-actin as an internal control. (**B**) Transferase-mediated dUTP-fluorescein nick end labeling (TUNEL) is used to test for DNA fragmentation. Three cell samples with average numbers of apoptotic positive cells are seen under the microscope (200× magnification). Densitometry was used to measure the difference in protein band intensities. The outcome was measured and expressed as a fold change compared to the control (*n* = 3). (**C**) Cells were pretreated with NAC (2 mM) for 1 h, followed by treatment with MGN (20 μM for 24 h) and/or DOX (1 μM) to determine the level of Bcl2, cleaved caspase-3, and cleaved PARP. Data are represented as mean ± SEM. One-way ANOVA plus Tukey’s post hoc analysis analyzed the comparison between groups. * *p* < 0.05 reflects significant differences from the healthy control. # *p* < 0.05 reflects significant differences in contrast to DOX alone and MGN with DOX-treated groups.

## Data Availability

The data presented in this study are available in the article.

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
