# Peer review of "Mangiferin Inhibits Apoptosis in Doxorubicin-Induced Vascular Endothelial Cells via the Nrf2 Signaling Pathway"

_ijms, 2021, doi:10.3390/ijms22084259_

Round 1
Reviewer 1 Report
This study demonstrated the protective role of mangiferin (MGN), a plant-derived polyphenolic compound, against doxorubicin (DOX)-induced vascular endothelial cell apoptosis. The authors demonstrated that MGN ameliorates DOX-induced toxicity via activation of PI3K/Akt-Nrf2 axis. However the article have less novelty because Nrf2 activating property of this compound are already known. In addition, English language is too poor and difficult to understand, although most of the data looks reasonable. Extensive English language editing would be necessary.
Comments,
The author proposed that MGN activates Nrf2 through PI3K/Akt activation. However it is inaccurate because the lack of inhibitor or knockdown experiment.
Inappropriate hyphenations should be corrected throughout text.
“μmol” and “μM” means different thing, should be properly described.
Figure 5B looks incorrect. MGN decreases Nrf2?
Author Response
Reviewer 1
This study demonstrated the protective role of mangiferin (MGN), a plant-derived polyphenolic compound, against doxorubicin (DOX)-induced vascular endothelial cell apoptosis. The authors demonstrated that MGN ameliorates DOX-induced toxicity via activation of PI3K/Akt-Nrf2 axis. However the article have less novelty because Nrf2 activating property of this compound are already known. In addition, English language is too poor and difficult to understand, although most of the data looks reasonable. Extensive English language editing would be necessary.
Thank you for giving us the opportunity to submit a revised draft of our manuscript entitled “Mangiferin inhibit apoptosis in doxorubicin induced vascular endothelial cells via Nrf2 signaling pathway” [Manuscript ID: IJMS 1154230] to International Journal of Molecular Science.
We appreciate the time and effort that you and the reviewers have dedicated in providing valuable feedback on our manuscript. We are grateful to the reviewers for their insightful comments, which have improved the paper.
We have incorporated changes that reflect all the suggestions provided by the reviewers. All changes are highlighted in green in the revised manuscript. Please see below for our point-by point response to the reviewers’ comments. If any responses are unclear or you wish additional changes, please do not hesitate to let us know.
Comments,
- The author proposed that MGN activates Nrf2 through PI3K/Akt activation. However it is inaccurate because the lack of inhibitor or knockdown experiment.
Thank you for your comments. As suggested, in our revised manuscript, we have additionally done new experiment with PI3K/AKT inhibitor (LY294002) Figure 4.
- Inappropriate hyphenations should be corrected throughout text.
We made the correction as per your suggestion.
- “μmol” and “μM” means different thing, should be properly described.
We made the correction as per your suggestion.
- Figure 5B looks incorrect. MGN decreases Nrf2?
In our revised manuscript, clear image was added as suggested.
Thank you for all your valuable comments and questions, which allowed us to improve the quality of the manuscript.
Reviewer 2 Report
Numerous studies have shown the underlying mechanism of beneficial effects of mangiferin on cells, tissues or organs in several models of various diseases. The current manuscript has also shown that mangiferin treatment could provide effective protection against toxic injury in HUVECs treated with doxorubicin. The results are on the whole clear and concise but data tend to be fragmentary and difficult to find links between them. Below are some comments for improvement.
Major comments
1. Based on the results, the downstream protein of PI3K/Akt pathway involved in controlling Nrf2 activity, stability, nuclear import and export and interaction with Keap1 in this experiment is still unknown. Also, the results did not reveal that downstream of PI3K can be pathways that are either Akt-dependent or –independent.
1) To determine how changes in PI3K-Akt signaling affected antioxidants and detoxifying proteins such as HO-1, catalase and NQO1, HUVECs cells can be treated with PI3K inhibitors, followed by measurement of ARE activity, antioxidants and detoxifying proteins.
2) It is necessary to examine the effect of PI3K inhibition under conditions when Nrf2 is activated, which can be induced by treating cells by an ARE inducer or generating cells overexpressing Nrf2.
3) To examine whether Akt activation sufficient for Nrf2 activation, Nrf2 activity can be measured between cells transfected with empty vector and cells transfected with the activated Akt gene.
4) To investigate how PI3K inhibition could influence nuclear localization of Nrf2 in HUVECs, it is possible to perform subcellular fractionation immunoblots in HUVECs exposed to a PI3K inhibitor when treated with doxorubicin.
5) Rather, aberrant PI3K/Akt signaling may drive many of the molecular mechanism contributing to increased reactive oxygen species (ROS) levels through direct modulation mitochondrial bioenergetics and activation of NADPH oxidases, or indirectly through the production of ROS as a metabolic by-product. For example, ROS generated by Akt signaling triggered from doxorubicin may affect Nrf2-Keap1 complexes.
2. It may need more evidences which signaling pathways should be taken in relation to the effect that Nrf2 inhibits apoptosis. A previous study has shown that Nrf2 exerts potent anti-apoptotic effects through the activation of ERK1/2/ETS like protein-P70S6K-P90RSK signaling axis.
3. It has been demonstrated that mangiferin has pharmacological activities, including antioxidative, antiaging, antitumor, antibacterial, antiviral, immunomodulatory, antidiabetic, anti-apoptotic and analgesic effects. In actual, it is hard to find out a specific mechanism underlying the protective effects of mangiferin in this disease model.
Minor comments
1. It may be necessary to use the units consistently in both text and Figure 1 legend: μmol or μmol/mL
2. It would be better to make the figures well arranged.
Author Response
Reviewer 2
Thank you for giving us the opportunity to submit a revised draft of our manuscript entitled “Mangiferin inhibit apoptosis in doxorubicin induced vascular endothelial cells via Nrf2 signaling pathway” [Manuscript ID: IJMS 1154230] to International Journal of Molecular Science.
We appreciate the time and effort that you and the reviewers have dedicated to providing valuable feedback on our manuscript. We are grateful to the reviewers for their insightful comments, which have improved the paper.
We have incorporated changes that reflect all the suggestions provided by the reviewers. All changes are highlighted in green in the revised manuscript. Please see below for our point-by point response to the reviewers’ comments. If any responses are unclear or you wish to add additional changes, please do not hesitate to let us know.
Comments
- Based on the results, the downstream protein of PI3K/Akt pathway involved in controlling Nrf2 activity, stability, nuclear import and export and interaction with Keap1 in this experiment is still unknown. Also, the results did not reveal that downstream of PI3K can be pathways that are either Akt-dependent or independent.
Thank you for your valuable comments. The PI3K/Akt signaling pathway is commonly involved in the Nrf2-dependent transcription in diverse cell types of responding ROS insults. In the present study cytoprotective effects of MGN at least partly depend on the PI3K/Akt-mediated Nrf2 signaling pathway. Now we have done additional experiment in figure 4.
- To determine how changes in PI3K-Akt signaling affected antioxidants and detoxifying proteins such as HO-1, catalase and NQO1, HUVECs cells can be treated with PI3K inhibitors, followed by measurement of ARE activity, antioxidants and detoxifying proteins. It is necessary to examine the effect of PI3K inhibition under conditions when Nrf2 is activated, which can be induced by treating cells by an ARE inducer or generating cells overexpressing Nrf2.
Thank you, I agree with the reviewer's point. As per your suggestion, we have done additional experiment in figure 4. We have GST and SOD abs from cell signaling, for others antioxidants unfortunately, we do not have stock. I have ordered, but the suppliers said there is no guarantees due to corona pandemic. I am very sorry; I will consider your suggestions in future. Thank you.
- To examine whether Akt activation sufficient for Nrf2 activation, Nrf2 activity can be measured between cells transfected with empty vector and cells transfected with the activated Akt gene.
Thank you for the reviewer's careful review. As per your suggestion, this has been done in my revised manuscript (figure 6C).
- To investigate how PI3K inhibition could influence nuclear localization of Nrf2 in HUVECs, it is possible to perform subcellular fractionation immunoblots in HUVECs exposed to a PI3K inhibitor when treated with doxorubicin.
Thank you for the reviewer's careful review. As you had suggested, this has been done in my revised manuscript (figure 4).
- Rather, aberrant PI3K/Akt signaling may drive many of the molecular mechanism contributing to increased reactive oxygen species (ROS) levels through direct modulation mitochondrial bioenergetics and activation of NADPH oxidases, or indirectly through the production of ROS as a metabolic by-product. For example, ROS generated by Akt signaling triggered from doxorubicin may affect Nrf2-Keap1 complexes.
We have to plan the effect of DOX induced oxidative stress via direct modulation or indirectly, to produce vascular damages, in some other normal endothelial cells HUVCEs, HAPEC and HCAEC. We will consider your suggestions in future. Thank you.
- It may need more evidences which signaling pathways should be taken in relation to the effect that Nrf2 inhibits apoptosis. A previous study has shown that Nrf2 exerts potent anti-apoptotic effects through the activation of ERK1/2/ETS like protein-P70S6K-P90RSK signaling axis.
Thank you, I agree with the reviewer's point. In our lab we have NAC inhibitor (N- acetyl cysteine). In our revised manuscript we have done an additional experiment by using this inhibitor and determined the expression of apoptotic related proteins.
- It has been demonstrated that mangiferin has pharmacological activities, including antioxidative, antiaging, antitumor, antibacterial, antiviral, immunomodulatory, antidiabetic, anti-apoptotic and analgesic effects. In actual, it is hard to find out a specific mechanism underlying the protective effects of mangiferin in this disease model.
As per your suggestions, our revised manuscript to found the specific mechanism of MGN against DOX induced oxidative stress in endothelium.
Minor comments
It may be necessary to use the units consistently in both text and Figure 1 legend: μmol or μmol/mL
Done as you suggested
It would be better to make the figures well arranged
Done as you suggested
Thank you for all your valuable comments and questions, which allowed us to improve the quality of the manuscript.
Round 2
Reviewer 1 Report
Thank you for extensive revisions and polite answers for reviewer’s comments. The authors added several important experimental data that are required for the conclusion of this study.
However, several points described below were not revised properly.
Comments
1) The reviewer strongly recommends to use English editing service or editing by a native speaker to improve the manuscript.
2) As suggested in the first round review, “μmol” and “μM” are different meaning.
The “mol (or mmol, μmol)” means amount of substance. See “https://en.wikipedia.org/wiki/Mole_(unit)”
On the other hand, the “M (or mM, μM)” means concentration of substance. (M = mol/L).
See “https://en.wikipedia.org/wiki/Molar_concentration”
So, “μmol” should be incorrect. It should be “μM” or 1”μmol/L” throughout the manuscript.
Line 280. “μmol/mL” should be “μmol/L” or “μM”.
Line 283, “μmol/mL” should be “μmol/L” or “μM”.
Line 298, “MGN 20 mol” should be “MGN 20 μM”.
Line 323, “10 and 20μmol/ml” should be “10 and 20μmol/l”
Line 352, “NAC (20 mmol)” should be “NAC (20 mM).
3) There are several typos such as
Line 24, PI3
Line 40 What is “FAS/FAS ligand transcription factor”?
Line 43, Phosphoinositiden-3-Kinase
Line 46, NQO1 and HO-1 are not phase II detoxifying proteins. Phase II detoxifying proteins are drug metabolizing enzymes, such as glutathione-S-transferases.
Line 47, “the electrophile response elements” is alternative name of ARE.
Line 234, Phosphatylinositol 3-PIK3-kinase
Line 532, conflu-ence
Line 533, cul-ture
Line 531, 5 x 104 cells
Line 540, 2 x 104 cells/mL
etc.
Author Response
Comments
- The reviewer strongly recommends to use English editing service or editing by a native speaker to improve the manuscript.
Our revised manuscript corrected with MDPI English editing (ID english-28718 ), (attached the certificate)
2) As suggested in the first round review, “μmol” and “μM” are different meaning.
The “mol (or mmol, μmol)” means amount of substance. See “https://en.wikipedia.org/wiki/Mole_(unit)”
On the other hand, the “M (or mM, μM)” means concentration of substance. (M = mol/L).
See “https://en.wikipedia.org/wiki/Molar_concentration”
So, “μmol” should be incorrect. It should be “μM” or 1”μmol/L” throughout the manuscript.
Line 280. “μmol/mL” should be “μmol/L” or “μM”.
Thank you for comments, done as you suggested
Line 283, “μmol/mL” should be “μmol/L” or “μM”.
Thank you for comments, done as you suggested
Line 298, “MGN 20 mol” should be “MGN 20 μM”.
Thank you for comments, done as you suggested
Line 323, “10 and 20μmol/ml” should be “10 and 20μmol/l”
Thank you for comments, done as you suggested
Line 352, “NAC (20 mmol)” should be “NAC (20 mM).
Thank you for comments, done as you suggested
3) There are several typos such as
Line 24, PI3
Thank you for comments, done as you suggested
Line 40 What is “FAS/FAS ligand transcription factor”?
Thank you for comments, done as you suggested
Line 43, Phosphoinositiden-3-Kinase
Thank you for comments, done as you suggested
Line 46, NQO1 and HO-1 are not phase II detoxifying proteins. Phase II detoxifying proteins are drug metabolizing enzymes, such as glutathione-S-transferases.
Thank you for comments, done as you suggested
Line 47, “the electrophile response elements” is alternative name of ARE.
Thank you for comments, done as you suggested
Line 234, Phosphatylinositol 3-PIK3-kinase
Thank you for comments, done as you suggested
Line 532, conflu-ence
Thank you for comments, done as you suggested
Line 533, cul-ture
Thank you for comments, done as you suggested
Line 531, 5 x 104 cells
Thank you for comments, done as you suggested
Line 540, 2 x 104 cells/mL
Thank you for comments, done as you suggested

Reviewer 2 Report
The authors tried to cater to all needs from the reviewer. The alphabets within the images should be in the same way, position or arrangement.
Author Response
The authors tried to cater to all needs from the reviewer. The alphabets within the images should be in the same way, position or arrangement.
Thank you for your comments, done as you suggested

Round 3
Reviewer 1 Report
As the reviewer suggested, "μmol" in the text are revised to "μM".
However, unfortunately "μmol" in the Figures are left as is.
Please carefully revise the manuscript as much as possible.